# *Aggregatibacter actinomycetemcomitans* LtxA Hijacks Endocytic Trafficking Pathways in Human Lymphocytes

**DOI:** 10.3390/pathogens9020074

**Published:** 2020-01-21

**Authors:** Edward T Lally, Kathleen Boesze-Battaglia, Anuradha Dhingra, Nestor M Gomez, Jinery Lora, Claire H Mitchell, Alexander Giannakakis, Syed A Fahim, Roland Benz, Nataliya Balashova

**Affiliations:** 1Department of Basic and Translational Sciences, School of Dental Medicine, University of Pennsylvania, Philadelphia, PA 19104, USA; kslpt@verizon.net (E.T.L.); battagli@upenn.edu (K.B.-B.); dhingra@upenn.edu (A.D.); tuk53759@temple.edu (N.M.G.); jlora@upenn.edu (J.L.); chm@upenn.edu (C.H.M.); alek@sas.upenn.edu (A.G.); razafahim1991@gmail.com (S.A.F.); 2Department of Life Science and Chemistry, Jacobs University Bremen, 28759 Bremen, Germany; r.benz@jacobs-university.de

**Keywords:** *Aggregatibacter actinomycetemcomitans*, RTX toxin, localized aggressive periodontitis, LFA-1, leukotoxin (LtxA), endocytosis

## Abstract

Leukotoxin (LtxA), from oral pathogen *Aggregatibacter actinomycetemcomitans*, is a secreted membrane-damaging protein. LtxA is internalized by β2 integrin LFA-1 (CD11a/CD18)-expressing leukocytes and ultimately causes cell death; however, toxin localization in the host cell is poorly understood and these studies fill this void. We investigated LtxA trafficking using multi-fluor confocal imaging, flow cytometry and Rab5a knockdown in human T lymphocyte Jurkat cells. Planar lipid bilayers were used to characterize LtxA pore-forming activity at different pHs. Our results demonstrate that the LtxA/LFA-1 complex gains access to the cytosol of Jurkat cells without evidence of plasma membrane damage, utilizing dynamin-dependent and presumably clathrin-independent mechanisms. Upon internalization, LtxA follows the LFA-1 endocytic trafficking pathways, as identified by co-localization experiments with endosomal and lysosomal markers (Rab5, Rab11A, Rab7, and Lamp1) and CD11a. Knockdown of Rab5a resulted in the loss of susceptibility of Jurkat cells to LtxA cytotoxicity, suggesting that late events of LtxA endocytic trafficking are required for toxicity. Toxin trafficking via the degradative endocytic pathway may culminate in the delivery of the protein to lysosomes or its accumulation in Rab11A-dependent recycling endosomes. The ability of LtxA to form pores at acidic pH may result in permeabilization of the endosomal and lysosomal membranes.

## 1. Introduction

The RTX (Repeats in ToXin) toxins are membrane-damaging proteins secreted by some Gram-negative bacteria [1]. The organisms producing these proteins are important human and animal pathogens, implicating the toxins’ role in the bacterial virulence. RTX toxins’ common features are using the type I secretion system as a mode of export across the bacterial envelope, employing an uncleaved C-terminal recognition signal [2,3,4], and the characteristic nonapeptide glycine- and aspartate-rich repeat binding of Ca^2+^ ions [5,6]. The toxins are modified with fatty acid moieties attached to internal lysine residues, which is an unusual characteristic for bacterial proteins [7,8,9,10]. RTX toxins can be divided into three groups: (i) broadly cytolytic RTX hemolysins, (ii) species-specific RTX leukotoxins, and (iii) large, multifunctional, autoprocessing RTX toxins (MARTX) [1]. RTX leukotoxins exhibit a narrow cell type and species specificity due to cell-specific binding through protein receptors of the β_2_ integrin family [1]. The β_2_ integrins are expressed on the surface of leukocytes and share a common β_2_ subunit, CD18, which is combined with either one of the unique α chains, α_L_ (CD11a), α_M_ (CD11b), α_X_ (CD11c), or α_D_ (CD11d) [11].

*Aggregatibacter actinomycetemcomitans* (*Aa*), a facultative anaerobe and common inhabitant of the human aerodigestive tract, causes localized aggressive periodontitis (LAP) [12]. LAP is a rapidly progressing periodontal disease that results in loss of tooth attachment and alveolar bone destruction in adolescents. If left untreated in teenagers, the infection will lead to the loss of the permanent first molars and central incisors [13]. Recent data indicate that *Aa* plays a role in the early stages of the disease. Specific *Aa* virulence factors can trigger the disease by suppressing the host response, which will allow for the overgrowth of *Aa* and other “toxic” bacteria in the local environment [12]. The pivotal virulence factor of *Aa* is an RTX leukotoxin, LtxA, that kills both human innate and adaptive immune cells [14]. *Aa* isolated from LAP patients predominantly belongs to a single clone, JP2 [15], which is characterized by increased LtxA production, implicating a role for LtxA in disease development [16]. Analysis of a primary LtxA sequence consisting of 1055 amino acids predicts four LtxA domains [17]. The hydrophobic domain encompasses residues 1–420 and incorporates cholesterol recognition amino acid consensus (CRAC) [18]. CRAC motif mediates LtxA binding to cholesterol and is essential for LtxA association with the plasma membrane of human T lymphocytes and monocytes [18,19]. The central domain (residues 421–730) contains two internal lysine residues (K^562^ and K^687^) that are the sites of post-translational acylation, required for LtxA activation [9]. The repeat domain (residues 731–900) contains the typical repeated amino acid sequence of the RTX family with the C-terminal domain (residues 901–1055), and is believed to play a role in secretion [17].

Recent findings suggest that recirculating and resident memory T cells in gingival tissue play an important role in the maintenance of periodontal homeostasis [20]. In an experimental rat periodontal disease model, antigen-specific CD4 T lymphocytes were required for bone resorption [21]. Hence, the investigation of the LtxA effect on T lymphocytes is important for our understanding of how *Aa* causes periodontal disease. In our previous studies, the Jurkat cell line, subclone Jn.9, served as a model to study LtxA interaction with T lymphocytes’ cell membrane. Jn.9 cells express cell-surface LFA-1 and are susceptible to LtxA-induced toxicity [22]. LtxA toxicity requires the presence of the β_2_ integrin LFA-1(LFA-1, CD11a/CD18 or α_L_/β_2_) and cholesterol on the surface of Jn.9 cells [18,23,24]. LFA-1 is a native ligand for intercellular adhesion molecule (ICAM-1) located on vascular endothelial cells [25]. In immunocytes, LFA-1/ICAM-1 binding is one of the molecular mechanisms for leukocyte adhesion and migration to the site of infection [26]. LFA-1 is constantly endocytosed and then rapidly recycled back to the plasma membrane through vesicular transport [27,28] using the “long-loop” of recycling involving GTPase Rab11A-positive endosomes [29]. Also, LFA-1 activity is regulated by the ability of these receptors to switch between active and inactive conformations [25]. 

In the proposed mechanism of LtxA interaction with the Jn.9 cells membrane, the initial binding of the toxin with the membrane elevates cytosolic Ca^2+^ independent of the toxin binding to LFA-1. Ca^2+^ elevation involves the activation of calpain and talin cleavage, and the subsequent clustering of LFA-1 in lipid rafts on the membrane [24]. LtxA binds to the extracellular domains of LFA-1 subunits, CD11a and CD18. The toxin then transverses the cell membrane, binds to the cytoplasmic tails of LFA-1, and causes activation of LFA-1 [23]. Following from the results of the liposomal study, LtxA adopts a U-shaped conformation in the membrane, with the N- and C-terminal domains residing outside of the membrane [30]. 

After binding to the LFA-1 subunits, LtxA is quickly internalized into the cytosol, where it is found in vesicular structures [23]. Since LtxA binds to LFA-1, there is a possibility that LtxA could be using an integrin endocytic trafficking pathway to gain access to the target cell cytosol. The mechanism of LtxA uptake and the pathway of intracellular toxin trafficking has not been investigated. Here, we examined the components of the cytosol of LtxA-treated cells for co-localization of the toxin and the CD11a subunit of LFA-1 with different organelle markers. LtxA association with endosomal and lysosomal markers suggests a receptor-mediated endocytic process that may culminate in the delivery of the toxin to lysosomes. Additionally, the toxin can be redirected to the plasma membrane due to the LFA-1 receptor Rab11A-mediated recycling. This study provides new insight into convergent mechanisms of LFA-1 and LtxA trafficking, and the ability of LtxA to function in acidic environments.

## 2. Results

### 2.1. LtxA Does Not Damage Host Cell Membrane When Entering the Cell

The membrane damaging properties of LtxA have been documented [31,32]. Therefore, the first question we asked was whether the initial steps of LtxA interaction with the cells result in the plasma membrane damage. The green-fluorescent impermeable nucleic acid stain YO-PRO®-1 is used to detect early membrane damage as it permeates cells immediately after membrane destabilization [33]. Propidium iodide (PI) is used to identify late cell death in which the integrity of the plasma and nuclear membranes significantly decreases, allowing PI to penetrate the membranes and intercalate into nucleic acids [34]. To study the effect on membrane permeability, we first incubated Jn.9 cells with 20 nM LtxA at different times over a 10 h time period, and then flow cytometry analysis of LtxA-treated cells was performed to determine YO-PRO®-1 and PI internalization. The YO-PRO®-1 membrane permeabilization assay showed no evidence of plasma membrane damage in LtxA-treated Jn.9 cells at least within first 3 h of treatment (Figure 1A,B). However, our flow cytometry data demonstrated that 20 nM LtxA-DY488 became internalized with Jn.9 cells 30 min after the toxin was added and internalization steadily increased over time (Figure 1A,B). Lymphocytes are known to be moderately susceptible to LtxA and are killed by apoptosis [35]. The staining of Jn.9 cells with PI was observed after 10 h of treatment with LtxA, suggesting that cells are in the late apoptotic stage (Figure 1B). Hence, our data indicate that LtxA is quickly internalized by Jn.9 cells but the toxin does not rupture the plasma membrane when it enters the host cells.

### 2.2. LtxA Uptake is Diminished by Dynamin Inhibitors

We hypothesized that LtxA could get access to the cytosol of Jn.9 cells through endocytic uptake. The exposure of cells to cold temperatures is used for the nonspecific inhibition of endocytosis [36]. Therefore, we treated Jn.9 cells with fluorescent-labeled LtxA in ice-cold medium and analyzed the toxin entry to cells by flow cytometry and confocal microscopy. We demonstrated that cold temperatures affected the binding of and slowed down internalization of the toxin with Jn.9 cells (Figure 2A, Appendix A). We then set up an experiment employing a set of chemical and pharmacological inhibitors of endocytosis (Table 1) to define the mechanism of the toxin uptake by the cells. The effect of dynamin- and clathrin-mediated endocytosis inhibitors on transferrin uptake is well established [37,38]. To confirm the efficiency and select the inhibitors’ concentrations for our study, the internalization of transferrin conjugated to Alexa Fluor®555 and LtxA-DY650 by Jn.9 cells was followed using confocal microscopy Appendix A. At 10 µM Dynasore, 10 µM Dynole 34-2 and 5 µM Pitstop 2, the inhibitory effect on transferrin uptake was observed. First, we wanted to identify whether GTPase dynamin activity is essential for LtxA uptake. The LtxA-DY650 internalization in the presence of dynamin inhibitor 10 µM Dynasore was evaluated by live confocal imaging (Appendix A) and flow cytometry analysis (Figure 2C). To confirm our results, we performed flow cytometry analysis in cells treated with another dynamin inhibitor, 10 µM Dynole 34-2, which blocks the GTPase activity of dynamin [37,39]. Fluorescent-labeled toxin internalization was significantly reduced in cells pre-treated with dynamin-inhibitors. Cells pretreated with 10 µM Dynasore and Dynole 34-2 for 20 min internalized much less LtxA (Figure 2C). However, the inactive control for Dynole 34-2, Dynole 31-2, did not inhibit the toxin internalization. The inhibitors affecting the clathrin-mediated endocytic pathway, such as potassium-depleted medium [40] and 5 µM Pitstop 2 [41], did not change the efficiency of LtxA internalization with Jn.9 cells (Figure 2B,C). Collectively, these data suggest that LtxA internalization with Jn.9 cells is dynamin-dependent and predominantly clathrin-independent. Next, we evaluated the toxicity of LtxA on Jn.9 cells containing the above inhibitors. However, we identified some toxic effect of endocytosis inhibitors on Jn.9 cells. After 18 h of treatment with the 2 nM toxin, we observed some decrease in LtxA toxicity on Jn.9 cells pretreated with 10 µM Dynasore and 10 µM Dynole 34-2 (Appendix A), suggesting that these compounds could attenuate LtxA intoxication.

### 2.3. LtxA and CD11a Are Found in Early and Recycling Endosomes

In our imaging studies, LtxA was found in vesicular structures after entry into Jn.9 cells [23]. The co-distribution of LtxA and LFA-1 heterodimer components on the surface of target cell membranes indicates that LtxA could intrude into the cytosol as individual LtxA molecules or as part of an LtxA/LFA-1 complex. In order to characterize LtxA-containing endocytic vesicles, Jn.9 cells were treated with fluorescent-labeled LtxA for 30 min and were used to perform immunocytochemistry experiments with endocytic pathway markers, including GTPase Rab5 and Rab11A. Our imaging studies demonstrated the abundant colocalization of LtxA, early endosome membrane protein Rab5 and CD11a, suggesting toxin uptake through receptor-mediated endocytosis. Figure 3 and Appendix A show confocal images of Jn.9 cells with a co-localization of LtxA, CD11a, and Rab5 after treatment of the cells with LtxA-DY650 for 30 min at 37 °C. 

LFA-1 is exocytosed via GTPase Rab11A-mediated recycling [43] a process that involves trafficking through the perinuclear recycling compartment (PNRC), before reaching the plasma membrane. We found co-localization of CD11a and LtxA with Rab11A, a marker of recycling endosomes in approximately 1/3 of LtxA-containing spots (Figure 4 and Appendix A). The interaction of CD11a and LtxA with Rab11A in recycling suggests that after entering the early endosome a significant amount of LtxA is redirected back to the membrane in the LFA-1 recycling turnover. Alternatively, the release of LtxA into PNRC can provide access to the nuclear membrane for LtxA. Indeed, in our imaging studies we often observed the toxin surrounding nuclei (Appendix A).

### 2.4. LtxA and CD11a Are Found in Late Endosomes and Lysosomes

At later timepoints, Jn.9 cells treated with fluorescent-labeled LtxA were used in immunocytochemistry experiments with the endocytic pathway markers GTPases Rab7 and Lamp1. After 1 h of treatment with LtxA-DY650, LtxA was associated with the late endosome membrane protein Rab7 (Figure 5 and Appendix A). Colocalization was detected in approximately 1/10 of LtxA-containing spots. Colocalization of LtxA with lysosomal marker Lamp1 after 2 h of treatment with LtxA-DY650 indicated that the toxin trafficking culminates in its delivery to the lysosomes, where LtxA was found separated from CD11a (Figure 6 and Appendix A). 

### 2.5. Rab5 siRNA Knockdown limits LtxA Toxicity

Irrespective of routes of internalization, endocytic cargoes are trafficked to early endosomes, where Rab5 GTPases is the key player in subsequent trafficking events [44]. We investigated the impact of Rab5a downregulation on LtxA uptake and toxicity on cells (Figure 7). Western blot analysis 24 h after transfection with Rab5a siRNA confirmed that Rab5a was significantly downregulated (≥ 90%) in Jn.9 cells compare to scrambled siRNA transfected cells. When transfected cells were treated with 20 nM LtxA for 18 h, the toxic effect of the toxin on Rab5a downregulated cells was 30% less than on control cells (Figure 7A). Internalization of LtxA was analyzed by flow cytometry after 30 min of treatment with 20 nM LtxA-DY488. No significant variations in the amount of internal fluorescence were detected in cells transfected with Rab5a siRNA (mean channel fluorescence (MCF) 28.2 ± 0.6) and cells using scrambled siRNA (MCF 29.8 ± 0.6) (Figure 7B). Our results suggest that the abolishment of Rab5a function does not affect LtxA internalization, but affects cytotoxicity.

### 2.6. LtxA Causes Lysosomal Damage in Jn.9 Cells

We detected LtxA in Jn.9 lysosomes, and therefore we wanted to see whether LtxA was able to cause lysosomal damage in the cells. We have probed the effect of LtxA on lysosomal integrity in the cells using lysosomal dye, LysoTracker® Green DND-26. We followed changes in lysosomal properties of the cells after the addition of 20 nM LtxA to the cells by live cell confocal imaging. No changes in LysoTracker staining intensity were detected within the first 90 min of treatment and about 15% decrease in the intensity was identified in Jn.9 cells after 2 h of treatment (Figure 8A), which may indicate lysosomal damage due to lysosomal membrane permeabilization or lysosome alkalization. In order to assess if LtxA causes the rupture of lysosomes, we identified the intensity of Lamp1 staining in the presence and absence of toxin. Similar Lamp1 staining intensity in both conditions would indicate that this is likely because of a rise in pH in intact lysosomes. A decrease in Lamp1 intensity would suggest that change in lysosomal pH is due to lysosomal rupture (Figure 8B). 

### 2.7. LtxA is Active in Lipid Bilayer Membranes at a Low pH

Pore formation by LtxA was studied in detail at a neutral pH [32,46]. In lipid bilayer membranes formed by asolectin, LtxA forms cation-selective channels with a single-channel conductance of approximately 1.2 nS in 1 M KCl (pH 6.0) [46]. Since LtxA is found in endocytic vesicles, we asked whether LtxA is also able to form ion-permeable channels and damage membranes at an acidic pH. To address this, we performed lipid bilayer experiments with wildtype LtxA at different pH-values ranging from pH 3.5 to pH 10.0. LtxA formed ion-permeable channels in 1 M KCl solutions under all these conditions (pH 3.5, 4.7, 7.5, 8.5 and 10.0). However, because the membranes became very fragile at very low and very high pH values (3.7 and 10.0) it was not possible to record too many single-channel events under these conditions. At the other pH values, the membranes were rather stable, and a sufficient number of single-channel events could be recorded in the experiments. Figure 9 shows a single channel recording of LtxA in 1 M KCl, 10 mM MES-KOH, pH 4.7. The channel had a somewhat reduced lifetime at this pH as compared with a neutral pH [46]. Figure 9B,C shows a histogram obtained from 47 LtxA channels recorded under these conditions. A fit of the histogram with a Gaussian function yielded an average single-channel conductance of 1.1 ± 0.3 nS, somewhat smaller than that at pH 6.0 (G = 1.2 ± 0.3 nS) [46]. Again, we found that the single-channel distribution was quite broad, similar to the conditions at pH 6.0 (Figure 9B,C, Table 2). 

We also studied the effect of high pH values on channel formation, mediated by LtxA. Ion-permeable channels were also observed at these conditions. The average single channel conductances at pH 7.5, 8.5 and 10 are shown in Table 2. The influence of the aqueous pH was rather small on the conductance of the LtxA channel, despite a possible shift of the selectivity of the LtxA channel from being slightly cation selective at pH 6.0 to a higher selectivity for potassium ions over chloride. 

No residual fluorescence was detected in 0.1% Triton X-100 permeabilized cells after the trypan blue treatment. Untreated cells (blue or black) served as a negative control. Representative flow cytometry histograms are shown. 

## 3. Discussion

Leukocytes need to rapidly move from blood vessels to tissues upon inflammation or infection. A crucial mechanism regulating this process is the subcellular trafficking of adhesion molecules, primarily integrins [47]. Integrins undergo constant endo/exocytic turnover, necessary for the dynamic regulation of cell adhesion. Bacterial toxins have developed a number of schemes to cross the membrane in order to enter the cell. LtxA evolved the strategy to target specifically β_2_ integrin LFA-1 on leukocytes’ surface [22]. This binding is required for toxin internalization [23].

We here report that LtxA is delivered to the cytosol of Jn.9 cells through endocytic trafficking. Historically, endocytic pathways are classified as either clathrin-dependent or clathrin-independent. The large GTPase dynamin [48] is hypothesized to be directly involved in closing off endocytic vesicles from the plasma membrane. The key players in the formation of clathrin-coated vesicles are dynamin [48] and adaptor proteins [49]. The studies with *Mannheimia haemolytica* LktA, another RTX leukotoxin, show that LktA is internalized in a *dynamin*-*2* and *clathrin*-*dependent* manner [50]. The following LktA-trafficking events involve the toxin binding to the mitochondria and interaction with cyclophilin D, a mitochondrial chaperone protein, in bovine lymphoblastoid cells [51].

Our data indicate that LtxA enters Jn.9 cells using a clathrin-independent mechanism (or predominantly uses this pathway). Our results correlate with the finding that LFA-1 is internalized through a clathrin-independent, cholesterol-dependent pathway and this process is essential for cell migration [52]. In this scenario, non-clathrin-coated lipid raft microdomains form 50–100 nm flask-shaped vesicles in the plasma membrane regions rich in lipid rafts [53]. Lipid-raft dependent endocytosis was shown to be dynamin-dependent [54] and may involve caveolae formation. Thus, we hypothesize that LtxA/LFA-1 is endocytosed through caveolae-mediated endocytosis. 

Bacterial toxins often piggyback existing endocytic trafficking pathways [55,56] to deliver active proteins to subcellular targets. The small GTPases Rab are essential regulators of intracellular membrane trafficking and exist in an inactive GDP-bound form and an active GTP-bound form [57]. The co-localization experiments with Rab5, Rab7, Lamp1 revealed that LtxA can follow the degradation pathway process that culminates in the delivery of the toxin to lysosomes. Rab5 localizes to early endosomes where it is involved in the recruitment of Rab7 and the maturation of these compartments to late endosomes [58]. Impaired Rab5a function affects endo- and exocytosis rates and, conversely, Rab5 overexpression increases the release efficacy [59]. Therefore, the termination of Rab5 function blocks the movement of proteins downstream of the endocytic pathway. Downregulation of Rab5a decreased LtxA toxicity, suggesting that further toxin trafficking is required for intoxication by LtxA. LFA-1 was suggested to undergo endocytic recycling through the long-Rab11A-dependent pathway with a transitional step at PNRC [29]. Here we, for the first time, demonstrated CD11a localization in Rab11A-containing endocytic vesicles. Extensive colocalization of CD11a and Rab11A was found in Jn.9 cells which were not treated with LtxA (Appendix A). While some LtxA follows LFA-1 in its recycling turnover, a portion of LtxA is separated from LFA-1 and the toxin proceeds to late endosomes and lysosomes. A proposed model of LtxA trafficking in lymphocytes is shown in Figure 10.

Interaction between integrins and their β-integrin ligands typically leads to enhanced cell survival and several immunological changes [60,61]. Our experiment using cell impermeable dye, YO-PRO®-1, serves to demonstrate that LtxA gains access to the Jn.9 cell cytosol without evidence of plasma membrane damage. Our study and others suggest that LtxA could accumulate in the lysosomes and alter lysosomal pH [62,63]. Damage to lysosomes by LtxA in human and rat monocytes cells [62,64] and in human erythroleukemia cells [62] was reported. In our previous study, treatment with 100 ng/mL LtxA led to cytosol acidification in K562 cells expressing LFA-1, presumably due to the leakage of lysosomal content, as was identified using a pH-sensitive indicator pHrodo^®^. This process correlated with the disappearance of lysosomes in the cytosol, as determined by both acridine orange and LysoTracker® Red DND-99 staining. Similarly, using LysoTracker® Red DND-99 dye, lysosomal damage was detected in malignant monocytes (THP-1 cells) as early as 15 min after treatment with LtxA, and reached 70% after 2 h of treatment (unpublished data). In these cells, LtxA was shown to localize to the lysosome where it induces active cathepsin D release [64]. Here, we demonstrate that LtxA causes changes in lysosomal pH in T lymphocytes, however, to a leser extent. As the Lysotracker dye is sensitive to luminal pH, the decrease in the dye staining could have resulted from either a rise in lysosomal pH or a decrease in the number of lysosomes. To distinguish between these possibilities, cells were stained for lysosomal marker Lamp1. The clear decrease in Lamp1 staining, combined with the decrease in the Lysotracker signal, forms a strong argument that the toxin decreased the number of lysosomes. As the transcription factor EB (TFEB) feedback systems try to increase lysosomal biogenesis [65], the most likely explanation for this is that the toxin has ruptured the lysosomes and overridden the TFEB pathways. The pore-forming properties of LtxA are well established [32,46]. Therefore, we propose that LtxA can cause permeabilization of the lysosomal membrane, and possibly other intracellular organelles after the toxin is released from lysosomes.

LtxA was reported to cause different cellular responses leading to cell death in LFA-1-expressing cells. Kelk et al. reported that LtxA lyses healthy monocytes by the activation of inflammatory caspase 1 and causes release of IL-1b and IL-18. In contrast to myeloid cells, LtxA uses a “slow mode” of lymphocyte killing. The killing of malignant lymphocytes requires Fas receptors and caspase 8 in both T and B lymphocytes [66]. In B lymphocytes (JY cells), LtxA caused loss of the mitochondrial membrane potential, cytochrome c release, reactive oxygen species release, and activation of caspases 3,7,9 [24]. One possible explanation for the cell death mechanism induced by LtxA is the degree of lysosomal damage caused by the toxins in the cell. The extent of lysosomal rupture will determine morphological outcomes following lysosomal membrane permeabilization. Extensive lysosomal injury may lead to necrotic cell death, while less substantial damage to lysosomes may instigate several apoptotic pathways, which can be attenuated by the inhibition of lysosomal cathepsins [66,67,68,69].

The planar lipid bilayer assay is a highly sensitive method that allows the characterization of the membrane damaging activity of RTX-toxins in different physical conditions [70]. A current model proposes that RTX-toxins form cation-selective channels with a diameter of 0.6–2.6 nm in artificial membranes formed of lipid mixtures such as the asolectin/n-decane membrane [46]. It was demonstrated that the membrane-damaging activity of LtxA in artificial bilayers did not require the presence of the receptor [71]. In the endocytic pathway, subsequent acidification may initiate proteolysis and conformational changes, resulting in the ability of toxins and viruses to cross the endocytic vesicle membrane, since drugs that interfere with the endosomal pH are able to block the infection [72,73]. In this study, we used this method to observe and compare the pore formation of LtxA at different pH. We demonstrated that LtxA is functional in acidic pH found in endocytic vesicles and lysosomes, which may result in their damage. RTX toxins are intrinsically disordered proteins, therefore changes in pH may affect their secondary structure and consequently change their activity [74]. Further investigation is required to improve our understanding of the intracellular events leading to LtxA-induced cytolysis.

In conclusion, our results show that LtxA enters the cytosol of Jurkat cells without evidence of plasma membrane damage, utilizing receptor-mediated endocytic mechanisms. In our studies, colocolization between LtxA/CD11a was demonstrated on the plasma membrane [23] and in the early steps of LtxA endocytic trafficking. Our results suggest that LtxA can accompany LFA-1 in its recycling pathway; however, the toxin molecules can apparently dissociate from the receptor in an acidic environment of endocytic vesicles and independently follow the degradative pathway. LtxA delivery to the terminal point of this route results in the lysosomal membrane rupture.

## 4. Materials and Methods 

### 4.1. Antibodies and Chemicals

The following primary antibodies were used; CD11a Alexa Fluor^™^ 594 clone HI111 (Biolegend, San Diego, CA), rabbit polyclonal anti-Rab5, anti-Rab11A, anti-Rab7, or anti-Lamp1 antibody (Abcam, Cambridge, UK), anti-beta-actin antibody (AnaSpec, Fremont, CA) (1:1000), and anti-LtxA monoclonal antibody 107A3A3 [75] in hybridoma supernatants (1:10 dilution). The following secondary antibodies were used: goat anti-rabbit IgG Alexa Fluor^®^488 (1:1000); horseradish peroxidase (HRP)-conjugated goat anti-mouse IgG (Fc) or (HRP)-goat anti-rabbit (Pierce, Rockford, IL) (1:10,000). Transferrin labeled with Alexa Fluor®555 was from Invitrogen (Waltham, MA, USA). Dynamin inhibitor Dynole 34-2 and its inactive control, Dynole 31-2, were purchased from SigmaAldrich (St. Louis, MO), Dynasore and Pitstop 2 (Abcam, Cambridge, UK). The inhibitors were used in the following concentrations: 10 μM Dynole 34-2; 10 μM Dynole 31-2; 10 μM Dynasore; 5 μM Pitstop 2. 

### 4.2. Cell Culture

Jn.9, a subclone of Jurkat cells [76] was utilized in this study. The cells were cultivated in RPMI 1640 medium containing 10% FBS, 0.1 mM MEM non-essential amino acids, 1x MEM vitamin solution, and 2 mM L-glutamine, and 0.5 μg/mL gentamicin at 37 °C under 5% CO_2_.

### 4.3. LtxA Purification and Labeling

*Aa* strain JP2 [77] was grown on solid AAGM medium [78] for 48 h at 37 °C in 10% CO_2_ atmosphere. The colony was then inoculated in 1.5 L of liquid AAGM medium and the culture was incubated for 18 h. The toxin was purified from cell culture supernatants as described previously [79]. Purified LtxA was labeled with DyLight^™^ 650 (LtxA-DY650) or DyLight^™^ 488 (LtxA-DY488) using DyLight™ Amine-Reactive dyes (Pierce). The toxin was purified after labeling using a Zeba™ Spin Desalting column (40 K MWCO, Thermo Fisher™ Scientific), according to previously published protocol [23]. 

### 4.4. Immunofluorescence

For LtxA trafficking studies, 1 × 10^6^ of Jn.9 cells were incubated with 20 nM LtxA-DY650 for 15 min to 2 h at 37 °C in the growth medium. The cells were then washed with PBS, fixed with 2% paraformaldehyde for 10 min, washed twice with PBS, and permeabilized with 0.2% Triton X-100 for 20 min. The cells were subsequently blocked with 4% BSA for 30 min at 37 °C, incubated with primary antibody for 18 h at 4 °C, washed, and incubated with secondary antibody conjugated to Alexa Fluor 488 for 1 h at 37 °C. The nuclei were stained with 1 μg/ml Hoechst 33342 (Molecular Probes™, Eugene, OR) for 15 min at 37 °C. Samples were mounted in Cytoseal mounting medium (Electron Microscopy Sciences, Hatfield, PA) and images captured with a Nikon A1R laser scanning confocal microscope (Nikon Instruments Inc., Melville, NY ) with a PLAN APO VC 60 × water (NA 1.2) objective at 18 °C. Data were analyzed using Nikon Elements AR 4.30.01 software. For co-distribution analyses, the Pearson’s’ coefficient of 0.55 was used as a cut off and was identified using circular ROIs. A Z-stack series consisting of seventeen individual planes 0.33 µm apart were assembled in 3D animations using Nikon Elements AR 4.30.01 software. Maximum intensity projection and standard LUT adjustment were used for the images’ presentation.

For live imaging of LtxA uptake, Jn.9 cells were washed with in the serum free medium and were placed to attach for 20 min in ibiTreat 60 μ-dishes (Ibidi, Madison, WI) coated with poly-l-lysine (Sigma St. Louis, MO, USA). After floating cells were removed, the attached cells were pretreated with specific inhibitor, if necessary, and 20 nM LtxA-DY650 or 1 µM transferrin labeled with Alexa Fluor®555 was added. The cells were examined using a Nikon A1R laser scanning confocal microscope with a 60× water objective at different intervals of treatment at 37 °C. 

### 4.5. Inhibitors

Chemicals stocks were prepared in DMSO and were added in the 1 µl volume to 1 ml of cells. To measure LtxA internalization inhibition, Jn.9 cells (1 × 10^6^ cells) were pre-incubated with 5–10 µM inhibitors for 20 min in the serum free medium at 37 °C and then 20 nM LtxA-488 was added for 30 min. The effect of intracellular K^+^-depletion was evaluated using previously published protocol [80]. Pelleted Jn.9 cells (1 × 10^6^ cells) were incubated in 2 ml hypotonic medium (RPMI/water, 1:1) for 5 min, followed by incubation in isotonic K^+^-free buffer (50 mM Hepes and 100 mM NaCl at pH 7.4) for 40 min at 37 °C, and then 20 nM LtxA-488 was added for 30 min. The toxin internalization assay was performed as described in the “*Flow cytometry*” section. Live imaging of LtxA uptake was performed as described in the “*Immunofluorescence*” section. For cytotoxicity evaluation, the cells were treated with 2 nM LtxA and cell viability was evaluated as described in the “*Cytotoxicity assay*” section. The cells treated with specific inhibitors alone served as a control 

### 4.6. Flow Cytometry

YO-PRO®-1 and PI internalization was investigated using Membrane permeability/dead cell apoptosis kit (Invitrogen, Carlsbad, CA) according to the manufacture’s protocol. Jn.9 cells (1 × 10^6^) were treated with 20 nM LtxA at 37 °C in Jn.9 culture medium at indicated times, washed with PBS and then treated with 0.1 µM YO-PRO®-1 or 1.5 µM PI, followed by flow cytometry analysis. To detect internalized LtxA, Jn.9 cells (1 × 10^6^ cells) were incubated with 20 nM LtxA-DY488 for the specified time on ice or at 37 °C in Jn.9 culture medium, washed with PBS, and total cell-associated fluorescence was analyzed. To quench the extracellular fluorescence, LtxA-DY488-treated cells were incubated with 0.025% trypan blue (Sigma, St. Louis, MO) for 20 min as described previously [42,45]. To quench the intracellular fluorescence cells were permeabilized using 0.1% Triton X-100 (SigmaArdrich, St. Louis, MO) for 10 min and then subjected to 0.025% trypan blue treatment. Fluorescence was measured using a BD LSR II flow cytometer (BD Biosciences). Ten thousand events were recorded per sample, and MCF values were determined using WinList^™^7.0 software (Verity Software House). No residual fluorescence was detected in 0.1% Triton X-100 permeabilized cells after the trypan blue treatment. Samples that were not treated with LtxA or LtxA-DY488 served as a control. 

For lysosomal staining analysis, Jn.9 cells were incubated with 20 nM LtxA for 2 h at 37 °C in the growth medium. Then 100 nM LysoTracker^®^ Green DND-26 (Life Technologies, Carlsbad, CA, USA) was added to LtxA-treated and control cells for 15 min. 

### 4.7. Protein Analyses

The protein concentration was determined by absorption at 280 nm on A1 NanoDrop spectrophotometer (Thermo Fisher Scientific, Waltham, MA). Proteins were resolved on 4% to 20% SDS-PAGE and visualized by staining with GelCode blue stain reagent (Pierce, Rockford, IL). The Western blot analysis was performed as described previously [70]. 

### 4.8. siRNA

The validated Silencer® Select siRNA targeting human Rab5a (ID s11678) and Silencer^®^ Select Negative Control #2 siRNA (catalog# 4390846) were synthesized by Life Technology (Carlsbad, CA, USA). Jn.9 cells were transfected with lipofectamine 2000 (Life Technologies, Carlsbad, CA, USA) according to the manufacturer’s instructions. For each transfection, 5 µl of the 20 µM siRNA stocks were added to 400 µl of Jn.9 cells grown to 90% confluency. Rab5a levels in 1 x 10^6^ Jn.9 cells were confirmed by Western blot analysis 24 h after transfection. β-actin served as a loading control. 

### 4.9. Cytotoxicity Assay

For toxicity tests, 2–20 nM LtxA was added to 1 x 10^6^ Jn.9 cells in growth medium and incubated for 18 h at 37 °C. The cell membrane permeability was determined with trypan blue assay using Vi-cell Cell Viability Analyzer (Beckman Coulter, Miami, FL). All reactions were run in duplicate; the assay was performed three independent times. Untreated cells were used as controls. 

### 4.10. Planar Lipid Bilayers

Lipid bilayer measurements have been described previously in detail [81]. In short, A Teflon chamber, containing two 5 mL compartments connected by a small circular hole with a surface area of about 0.4 mm^2^, were filled with 1 M KCl, 10 mM MES, pH 6.0. Black lipid bilayer membranes were created by painting onto the hole solutions of 1% (w/v) asolectin (phospholipids from soybean, Sigma-Aldrich) in *n*-decane. The temperature was maintained at 20 °C during all experiments. The current across the membrane was measured with a pair of Ag/AgCl electrodes with salt bridges switched in series with a voltage source and current amplifier Keithley 427 (Keithley Instruments, INC. Cleveland, OH). The amplified signal was recorded by a strip chart recorder (Rikadenki Electronics GmbH, Freiburg, Germany). 

### 4.11. Statistical Analysis

The statistical analyses were performed using either Student’s test or one-way analysis of variance using SigmaPlot^®^ (Systat Software, Inc. Chicago, IL, USA). The following statistical criteria were applied: *p* < 0.001, *p* < 0.05, and *p* < 0.01.

## Figures and Tables

**Figure 1 pathogens-09-00074-f001:**
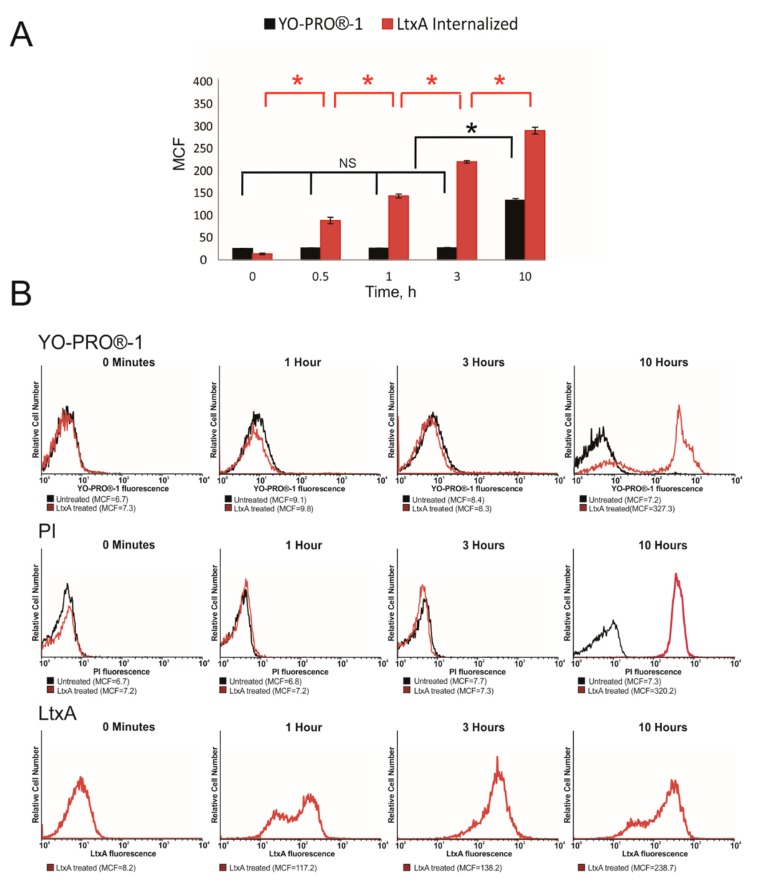
Damage to the plasma membrane in Jn.9 cells by LtxA. Flow cytometry analysis was used to detect YO-PRO®-1, PI and LtxA internalization with Jn.9 cells over time. Cells (1 × 10^6^) were incubated with 0.1 µM YO-PRO®-1/ 1.5 µM PI alone or after treatment with 20 nM LtxA at indicated times at 37 °C. Another set of cells was treated with 20 nM LtxA-DY488 at different times. The extracellular fluorescence of the cells was quenched with 0.025% trypan blue [23] and the intracellular fluorescence was determined. (**A**). Uptake of YO-PRO®-1 (black) and internalization of LtxA-DY488 (red) at various times, presented as mean channel fluorescence (MCF). The data shown are the results of three independent experiments. Error bars indicate ± SEM, * *p* ≤ 0.05. (**B**). Top and Middle: Flow cytometry histograms showing YO-PRO®-1 and PI dyes uptake by LtxA-treated cells (red line) vs. the dyes uptake by untreated Jn.9 cells (black line) at different times. Bottom: Flow cytometry histograms showing LtxA-DY488 internalized with Jn.9 cells. The data shown are representative of three independent experiments.

**Figure 2 pathogens-09-00074-f002:**
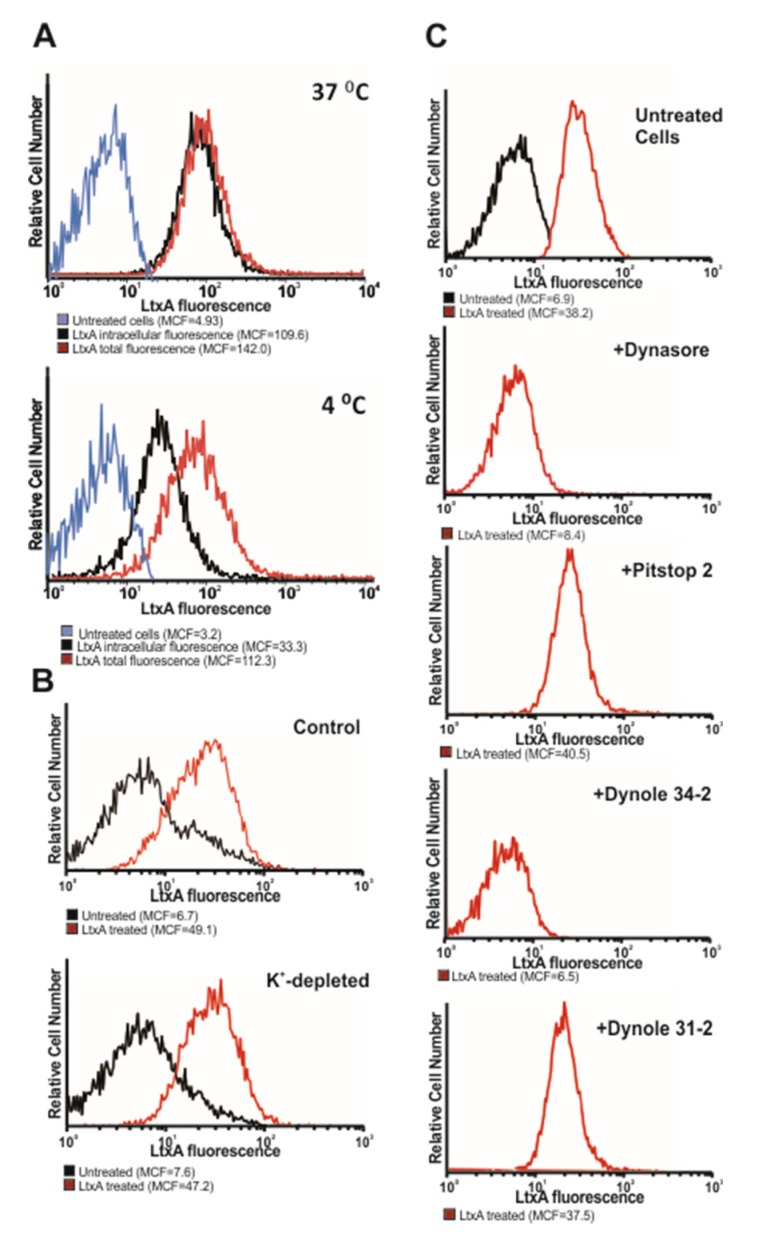
Effect of endocytosis inhibitors on LtxA internalization. (**A**). Flow cytometry analysis of LtxA internalization with Jn.9 cells at different temperatures. The cells were treated with LtxA-DY488 for 30 min on ice or 37 °C. In a set of cells, the total cell-associated fluorescence was measured by flow cytometry analysis (shown in red). In another set of cells, the extracellular fluorescence was quenched (0.025% trypan blue) [42] and intracellular fluorescence (red peak) was determined by flow cytometry analysis. (**B**). Flow cytometry analysis of LtxA internalization with Jn.9 cells in K^+^-free buffer. Jn.9 cells (1 × 10^6^) were incubated in K^+^-containing (top) or K^+^-free buffer (bottom), and then 20 nM LtxA-DY488 was added for 30 min. Flow cytometry analysis to determine the amount of internalized toxin (red peak) was performed as described in Figure 2A. (**C**). Flow cytometry analysis of LtxA-DY488 internalization with Jn.9 cells pretreated with chemical inhibitors. Jn.9 cells (1 × 10^6^) were preincubated with 5–10 µM endocytosis inhibitors for 20 min in serum free medium, and then were treated with 20 nM LtxA-DY488 for 30 min at 37 °C. The extracellular fluorescence was quenched 0.025% trypan blue, and intracellular fluorescence (red peak) was determined by flow cytometry analysis.

**Figure 3 pathogens-09-00074-f003:**
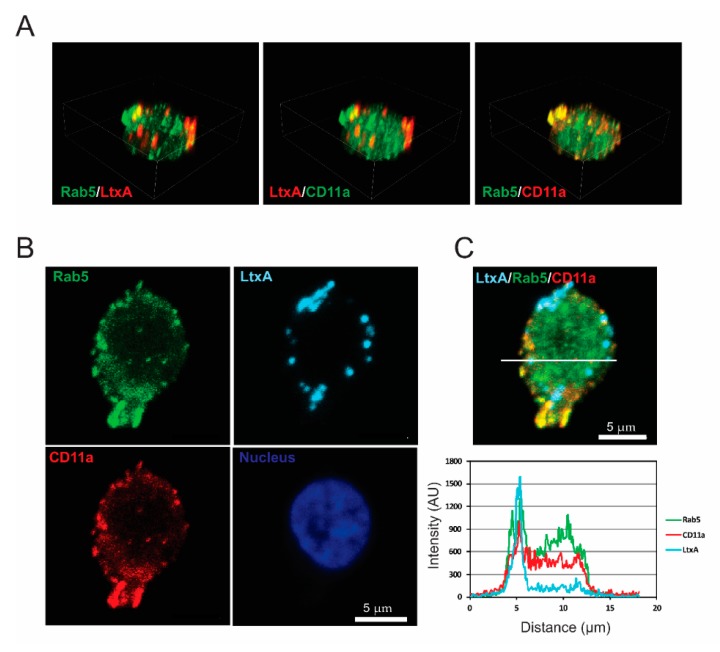
LtxA localization in early endosomes of Jn.9 cells. The cells were treated with 20 nM LtxA for 30 min at 37 °C. (**A**). 3D confocal images showing the distribution of LtxA, CD11a and Rab5. LtxA is pseudo colored in red and CD11a is in red or green (pseudo colored), as indicated on images. The 3D images were reconstructed from seventeen confocal planes using Nikon Elements AR 4.30.01 software. Bounding box dimensions are: width 14.19 µm; height 17.30 µm; depth 5.20 µm. (**B**). Localization of LtxA-DY650 is shown in cyan, CD11a recognized with mouse Alexa Fluor™ 594 clone HI111 is shown in red, and Rab5, recognized by rabbit anti-Rab5 antibody followed by staining with anti-rabbit IgG Alexa Fluor®488, is shown in green. The nucleus was stained with Hoechst dye and is shown in blue. (**C**). Top: Merged image “B” showing colocalization of LtxA DY650 (cyan), CD11a (red) and Rab5 (green). Bottom: Intensity profiles for LtxADY650 (cyan), CD11a (red) and Rab5 (green) across the line depicted in the image above. The degree of overlap in the LtxA-containing area was estimated with the Pearson’s correlation coefficient in the LtxA-containing area of 0.78 for LtxA and Rab5, 0.68 for LtxA and CD11a, and of 0.88 for CD11a and Rab5. Representative cells are shown. Additional data are shown in Appendix A.

**Figure 4 pathogens-09-00074-f004:**
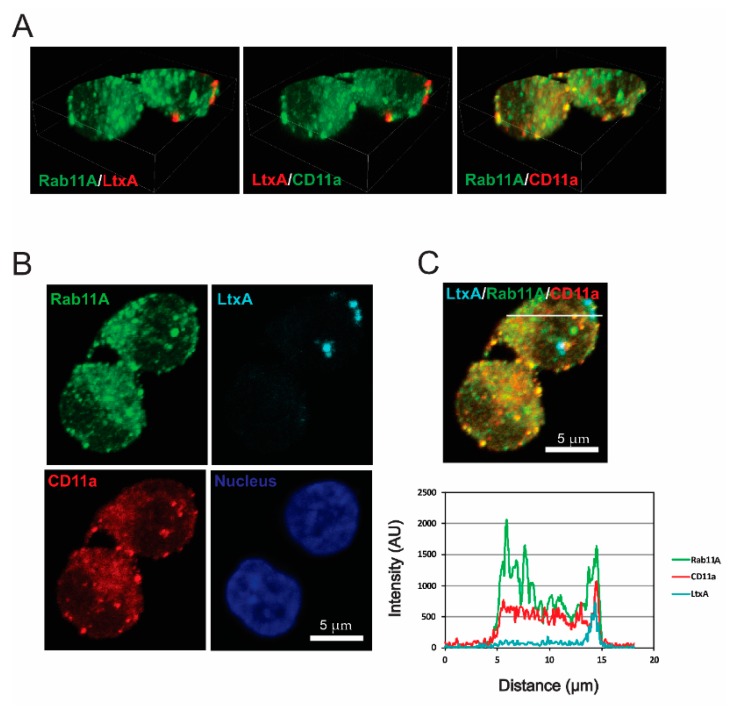
LtxA localization in Rab11A-positive endosomes of Jn.9 cells. The cells were treated with 20 nM LtxA for 30 min at 37 °C. (**A**). 3D confocal images showing the distribution of LtxA, CD11a and Rab11A. LtxA is pseudo colored in red and CD11a is in red or green (pseudo colored), as indicated on images. The 3D images were reconstructed from seventeen confocal planes using Nikon Elements AR 4.30.01 software. Bounding box dimensions are: width 22.58 µm; height 16.47 µm; depth 5.20 µm. (**B**). Localization of LtxA-DY650 is shown in cyan, CD11a recognized with mouse Alexa Fluor™ 594 clone HI111 is shown in red, and Rab11A, recognized by rabbit anti-Rab11A antibody followed by staining with anti-rabbit IgG Alexa Fluor®488, is shown in green. The nucleus was stained with Hoechst dye and is shown in blue. (**C**). Top: Merged image “B” showing co-localization of LtxA DY650 (cyan), CD11a (red) and Rab11A (green). Bottom: Intensity profiles for LtxADY650 (cyan), CD11a (red) and Rab11A (green) across the line depicted in the image above. The degree of overlap in the LtxA-containing area was estimated with the Pearson’s correlation coefficient of 0.75 for LtxA and Rab11A, 0.72 for LtxA and CD11a, and of 0.76 for CD11a and Rab11A. Representative cells are shown. Additional data are shown in Appendix A.

**Figure 5 pathogens-09-00074-f005:**
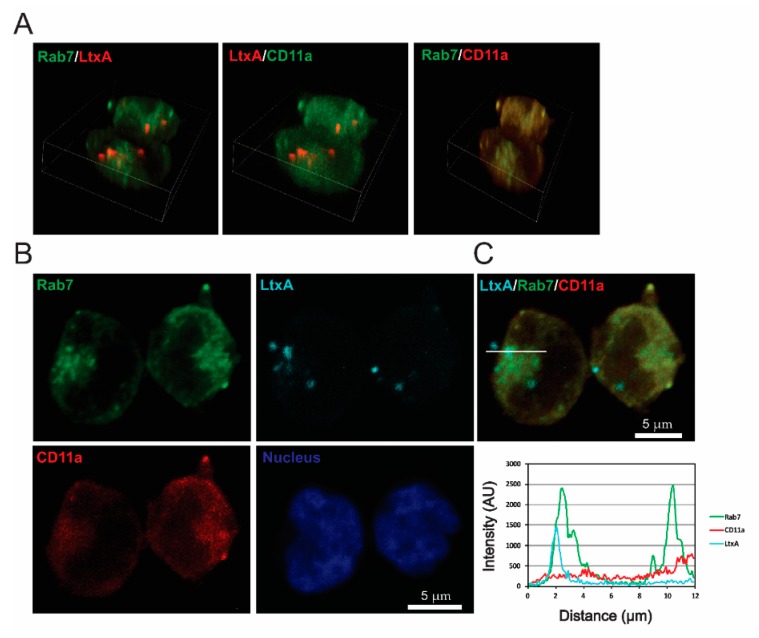
LtxA localization in Rab7-positive endosomes of Jn.9 cells. The cells were treated with 20 nM LtxA for 1 h at 37 °C. (**A**). 3D confocal images showing the distribution of LtxA, CD11a and Rab7. LtxA is pseudo colored in red and CD11a is in red or green (pseudo colored), as indicated on images. The 3D images were reconstructed from seventeen confocal planes using Nikon Elements AR 4.30.01 software. Bounding box dimensions are: width 22.17 µm; height 19.37 µm; depth 5.20 µm. (**B**). Localization of LtxA-DY650 is shown in cyan, CD11a recognized with mouse Alexa Fluor™ 594 clone HI111 is shown in red, and Rab7, recognized by rabbit anti-Rab7 antibody followed by staining with anti-rabbit IgG Alexa Fluor®488, is shown in green. The nucleus was stained with Hoechst dye and is shown in blue. (**C**). Top: Merged image “B” showing colocalization of LtxA DY650 (cyan), CD11a (red) and Rab7 (green). Bottom: Intensity profiles for LtxADY650 (cyan), CD11a (red) and Rab7 (green) across the line depicted in the image above. The degree of overlap in the LtxA-containing area was estimated with the Pearson’s correlation coefficient of 0.88 for LtxA and Rab7, 0.03 for LtxA and CD11a, and of 0.13 for CD11a and Rab7. Representative cells are shown. Additional data are shown in Appendix A.

**Figure 6 pathogens-09-00074-f006:**
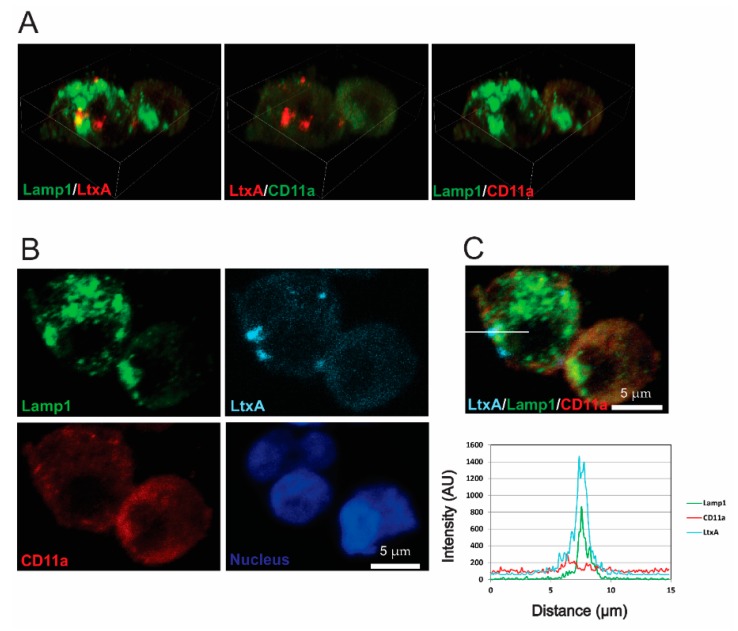
LtxA localization in lysosomes of Jn.9 cells. The cells were treated with 20 nM LtxA for 2 h at 37 °C. (**A**). 3D confocal images showing the distribution of LtxA, CD11a and Lamp1. LtxA is pseudo colored in red and CD11a is in red or green (pseudo colored), as indicated on images. The 3D images were reconstructed from seventeen confocal planes using Nikon Elements AR 4.30.01 software. Bounding box dimensions are: width 25.17 µm; height 19.37 µm; depth 5.20 µm. (**B**). Localization of LtxA-DY650 is shown in cyan, CD11a recognized with mouse Alexa Fluor™ 594 clone HI111 is shown in red, and Lamp1, recognized by rabbit anti-Lamp1 antibody followed by staining with anti-rabbit IgG Alexa Fluor®488, is shown in green. The nucleus was stained with Hoechst dye and is shown in blue. (**C**). Top: Merged image “B” showing colocalization of LtxA DY650 (cyan), CD11a (red) and Lamp1 (green). Bottom: Intensity profiles for LtxADY650 (cyan), CD11a (red) and Lamp1 (green) across the line depicted in the image above. The degree of overlap in the LtxA-containing area was estimated with the Pearson’s correlation coefficient of 0.72 for LtxA and Lamp1, 0.15 for LtxA and CD11a, and of 0.11 for CD11a and Lamp1. Representative cells are shown. Additional data are shown in Appendix A.

**Figure 7 pathogens-09-00074-f007:**
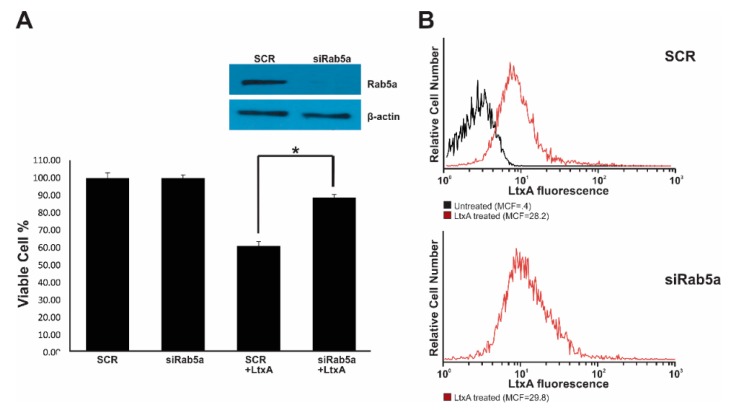
Modulation of Rab5a function in Jn.9 cells. A. Jn.9 cells (1 × 10^6^ cells) were transfected with siRNA control (SCR) or with siRNA against Rab5a and collected 24 h post-transfection for Rab5a expression analysis by Western blotting. The cell viability testing was performed by trypan blue assay after 18 h of treatment with 20 nM LtxA. (**A**). representative expression of Rab5a protein was shown for 24 h of siRNA treatment. The Rab5a protein expression (inset) was analyzed in extracts obtained from 1 × 10^6^ Jn.9 cells by Western blot, β-actin served as a loading control. Error bars indicate ±SEM, * *p* ≤ 0.05 compared with siRNA SCR-treated cells. The experiment was performed three independent times. (**B**). Jn.9 cells (1 × 10^6^ cells) were transfected with siRNA control (SCR) or with siRNA against Rab5a, then were collected 24 h post-transfection and treated with 20 nM LtxA-DY488 for 30 min at 37 °C. The extracellular fluorescence of the cells was quenched (0.025% trypan blue) [42,45] and intracellular cell fluorescence (red peak) was determined by flow cytometry analysis. No residual fluorescence was detected in 0.1% Triton X-100 permeabilized cells after the trypan blue treatment. Untreated cells (black) served as a negative control. Representative flow cytometry histograms are shown.

**Figure 8 pathogens-09-00074-f008:**
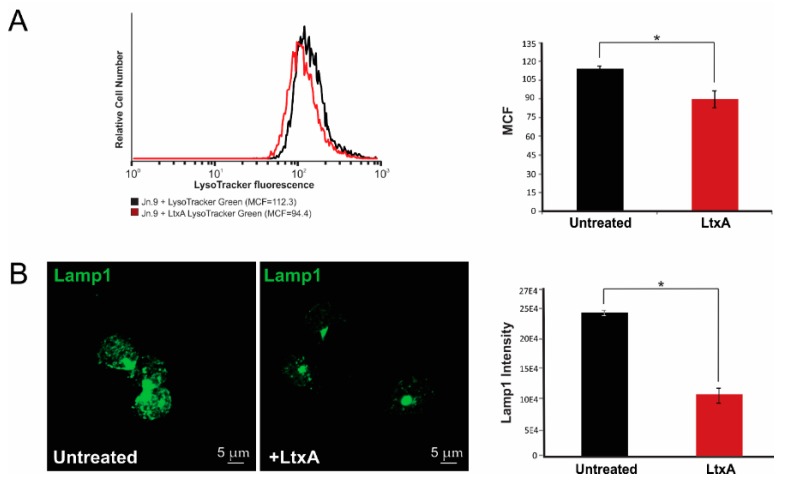
Lysosomal damage by LtxA. Jn.9 cells (1 × 10^6^) were incubated in presence or absence of 20 nM LtxA for 2 h at 37 °C. (**A**). LtxA-treated and untreated cells were stained with 100 nM LysoTracker^®^ Green DND-26 for 15 min at 37 °C. The LysoTracker^®^ Green DND-26 intensity was evaluated by flow cytometry analysis. Representative flow cytometry histograms are shown on the left. The fluorescence of LtxA treated cells is shown in red, and untreated cells in black. The mean channel fluorescence (MCF) of LysoTracker^®^ Green DND-26 in LtxA-treated vs. untreated cells is presented on the right. (**B**). Average fluorescence intensity of Lamp1 staining in Jn.9 cells was evaluated by confocal microscopy. The intensities of Lamp1 staining in 39 LtxA-treated cells and 44 untreated were analyzed and shown on the right. Error bars indicate ± SEM of three independent experiments. * *p* ≤ 0.05.

**Figure 9 pathogens-09-00074-f009:**
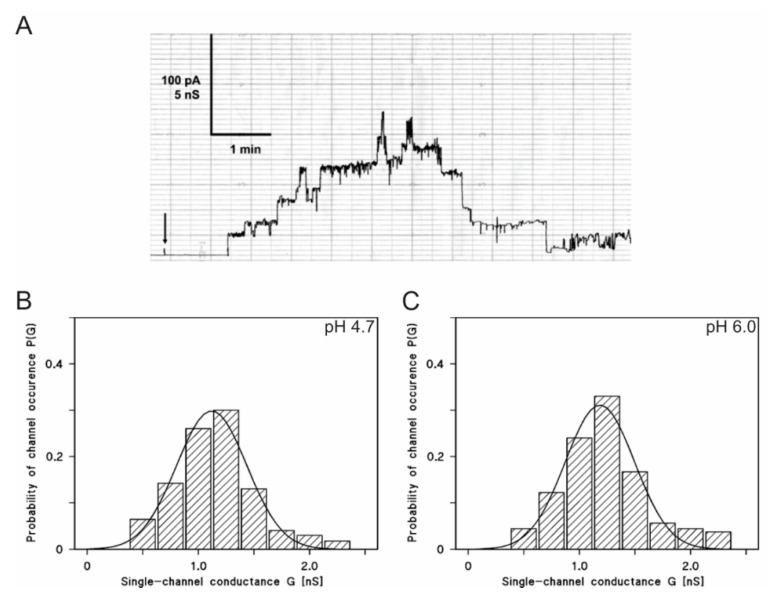
Pore-forming activity of LtxA in asolectin/*n*-decane membranes at different pH values. (**A**). Single-channel recording of LtxA in an asolectin/n-decane membrane at pH 4.7. Current recording of an asolectin/n-decane membrane, performed in the presence of 10 nM LtxA added to the cis-side of the membrane. The aqueous phase contained 1 M KCl, 10 mM MES-KOH, pH 4.7. The applied membrane potential was 20 mV at the cis-side (indicated by an arrow), at 20 °C. (**B**). Histogram of the probability P (G) of an occurrence of a given conductivity unit observed for LtxA with membranes formed of 1% asolectin dissolved in *n*-decane in a salt solution at pH 4.7. The histogram was calculated by dividing the number of fluctuations with a given conductance unit by the total number of conductance fluctuations. The average conductance was 1.1 ± 0.31 nS for 47 conductance steps derived from nine individual membranes. The value was calculated from a Gaussian distribution of all conductance fluctuations (solid line). The aqueous phase contained 1 M KCl, 10 mM MES-KOH, pH 4.7 and 10 nM LtxA; the applied membrane potential was 20 mV at 20 °C. (**C**). Histogram of the probability P(G) for the occurrence of a given conductivity unit observed for LtxA with membranes formed of 1% asolectin dissolved in n-decane in a salt solution at pH 6.0. The average conductance was 1.20 ± 0.31 nS for 95 conductance steps derived from 17 individual membranes. The aqueous phase contained 1 M KCl, 10 mM MES, pH 6.0 and about 10 nM LtxA; the applied membrane potential was 20 mV at 20 °C.

**Figure 10 pathogens-09-00074-f010:**
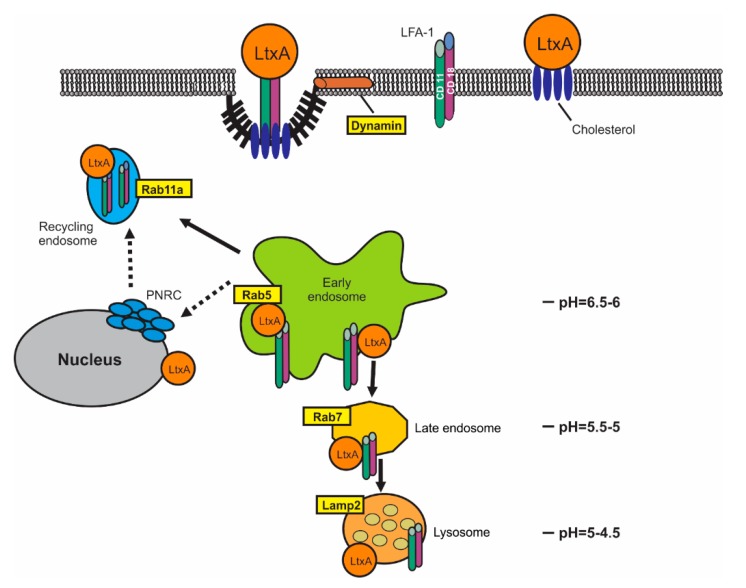
Proposed mechanism of LtxA entry and trafficking in human lymphocytes. LtxA binds to cholesterol and LFA-1 on the surface of Jn.9 cell. LtxA/LFA-1 complex internalization is dynamin-dependent. Internalized LtxA/LFA-1 complexes are quickly transported to early endosomes. The small GTPase Rab5 regulates membrane binding and fusion in the early endocytic pathway. The interruption of Rab5 expression in Jn.9 cells results in the abolishment of the LtxA activity. LFA-1 undergoes endocytic recycling through the long-Rab11A-dependent pathway with a transitional step at PNRC [29]. While some LtxA follows LFA-1 in its recycling turnover, a portion of LtxA is separated from LFA-1 and the toxin proceeds to late endosomes and lysosomes. The ability of LtxA to damage lipid membranes at a low pH may cause endocytic vesicles and lysosomal rupture and release of the toxin to the cytosol.

**Table 1 pathogens-09-00074-t001:** Chemical inhibition of LtxA uptake.

Compound	Mode of Action	Effect on Internalization
10 µM Dynasore *	Blocks GTPase activity of dynamin [37]	Inhibits
10 µM Dynole 34-2 *	Blocks GTPase activity of dynamin [39]	Inhibits
10 µM Dynole 31-2 *	Inactive derivative of Dynole 34-2	No effect
5 µM Pitstop 2 *	Interferes with binding of proteins to the N-terminal domain of clathrin [41]	No effect
K^+^-depletion	Inhibits clathrin mediated endocytosis [40]	No effect

* To measure LtxA internalization inhibition, Jn.9 cells were preincubated with 5–10 µM inhibitors for 20 min in serum free medium. The 0.5–1 mM chemical stocks were prepared in dimethyl sulfoxide (DMSO) and were added in the 1 µl volume to 1 ml of cells. No adverse effect of DMSO alone on Jn.9 cells was observed.

**Table 2 pathogens-09-00074-t002:** Influence of the aqueous pH on the conductance of channels formed by LtxA.

Salt and Buffer	pH	G * ± SD (nS)	Number of Events (n)
1 M KCl, 10 mM MES-KOH	3.7	1.0 ± 0.21	12
1 M KCl, 10 mM MES-KOH	4.7	1.1 ± 0.31	47
1 M KCl, 10 mM MES-KOH	6.0	1.2 ± 0.30	95
1 M KCl, 10 mM Tris-HCl	7.5	1.2 ± 0.24	39
1 M KCl, 10 mM Tris-HCl	8.5	1.3 ± 0.29	53
1 M KCl, 10 mM Tris-HCl	10	1.2 ± 0.26	14

* The LtxA conductance (G ± variance/SD) in each 1 M KCl solution was either taken from Gaussian distributions (see Figure 9) or directly from the statistics of single-channel data (n number of single events). To analyze the conductance in each case, n channels were reconstituted in asolectin/n-decane membranes at 20 mV voltage at 20 °C. The number of events analyzed at pH 3.7 and 10 was low due to the instability of the lipid bilayers at extreme pH values.

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
