# Peer review of "Aggregatibacter actinomycetemcomitans LtxA Hijacks Endocytic Trafficking Pathways in Human Lymphocytes"

_pathogens, 2020, doi:10.3390/pathogens9020074_

Round 1

Reviewer 1 Report

Results

Fig.1

-It should be interesting to show the effect of a LtxA dose-dependency curve on YOPRO/PI signal.

-Little has been commented in the text about the effect of the toxin at 10 hours.

Fig.1S

-They use the temperature control to show that the entry of the toxin is temperature-dependent. But in the way the assay has been performed, it is not discernible if the temperature is affecting the entrance or directly the binding itself. It should improve the assay to clarify this effect.

Fig.2, Fig.2S and Fig.3S

- The better inhibitor concentration should be chosen after carried out a dose-response assay. Especially, in this case, on clathrin-mediated endocytosis inhibitors.

-They have no tested the effect of CME inhibitors on transferrin internalization, as they did with dynasore. Why?

-Since the receptor enters accompanied by the toxin, it would be interesting to assess the effect of the toxin's inhibitors on receptor endocytosis.

-Inhibitors tested in this manuscript have been demonstrated that can have side-effects on other entry routes. It would be very recommended to include confirming experiments where inhibiting specifically the different entry routes using targeted siRNAs.

-In the Discussion section, they hypothesized that LtxA/CD11a complex is internalized through caveolae-mediated endocytosis. And for that they only show Fig9S. They should explore further this hypothesis and incorporate more data about it.

-It would be recommended to show data on Fig.2C as an independent supplementary figure. Nevertheless, some doubts arise whether this assay, as it is designed, is the most confident to show functional effects of the toxin's endocytosis inhibition.

Fig.3, Fig.4, Fig.4S, Fig.5, Fig.5S, Fig.7S

-The same cell/set of cells are shown in fig 1S, 3B and 4S.

-Although CD11a is described as the receptor of the toxin, except in one cell, there is little or no colocalization in the others. Also, the confocal microscopy images could show that the toxin promotes the internalization of the receptor (probably by its activation) and that the toxin presumably enters with it. What about that?

-According to the correlation data, there is the same correlation between LtxA-Rab5 in Fig. 3 and LtxA-Rab11 in Fig 4 (0.78), LtxA-CD11a in Fig3 and LtxA-CD11a in Fig4 (0.68), and CD11a-Rab5 in Fig3 and CD11a-Rab11 in Fig4 (0.88). This seems too coincidental, so it would be appropriate to review the data so as not to make mistakes.

-In Figure 4S and 5S, they should remove (cyan) and say (red / green) when it puts (red).

-Figure 7S is apparently not correct. The same fluorescence signal has been described for LtxA and CD11a. In addition, what is seen in Figure 5 does not correspond to what is seen in Figure 7S. The correlation coefficient calculated for Rab7-CD11a in Figure 5 is 0.13, while the image in Figure 7S shows a clearly higher correlation.

Fig.7

-On line 184, where it says Fig. 5A, it should say Fig. 7A.

-Data shown in Fig.7B are poor and not very convincing. It could be appreciable two incipient peaks in LtxA treated sample in siRab5a histogram

Figura 8

-On line 337 it should say nM instead of nm.

Figura 9

-The figure caption is somewhat repetitive

Figura 9S

-The figure corresponding to the abcam antibody used does not provide novel or relevant information, so its elimination is suggested.

-To study the co-localization with CD11a, 30 min incubation with 20 nM toxin is used. Why in this case a shorter time of only 15 min is used?

-More quantitative data is required for this figure to be relevant.

Figura 10S

- No quantitative co-localization data is given. They should be more cautious in affirming that for the first time they show the CD11a-Rab11a co-localization in endocytic vesicles.

MM

Inhibitors

-On line 515, change puffer to buffer

Flow cytometry

-I have doubts on how the toxin internalization quantification has been carried out. They state that they substract MCF values of trypan blue pretreated (¿?) cells to MCF values of total cell-asssociated fluorescence. If they did as the said, they were calculating cell-bound toxin fraction that is outside, not internalized fraction. The internalized fraction is the signal obtained after trypan blue is added. Could you explain better this point?

-For the LtxA internalization measurement and YO-PRO-1 and PI entry analysis in Fig.1, in which culture medium cells were incubated with the toxin?

-Typically, Trypan blue is used a higher concentration (0.2-0.4%) to quench outside 488 fluorescence signal in similar experiments. In this manuscript 0.025% concentration is used, as in the reference they include. Did you test if the Trypan blue concentration is enough to quench all the outside fluorescence and thus obtain only the internalized fraction?

Discussion

-They should study the hypothesis of the internalization of the LtxA mediated by caveolae in the results section.

-Although tentative, the argumentation given in relation to the pH acidification and the profound effects that it has on toxin's cytotoxic effects is poor and not convincing.

References

-References related to the methodological section are old. Their update is recommended.

Reviewer 2 Report

I want to congratulate to the authors for the extensive research conducted. 

However I have some suggestions to be addressed.

intro lines 49-50: unprecise: the disruption of the periodontal tissue due to the microbe is due to the unbalanced reply of the immune system to the microbial infection (https://www.ncbi.nlm.nih.gov/pubmed/31060232, https://www.ncbi.nlm.nih.gov/pubmed/31034083). 

Results 

take out any comments and move it into the discussion and therefore adjust the discussion: do not be too repetitive. 

I am missing a summarizing conclusion 

please add just to be more clear

Round 2

Reviewer 1 Report

Lines 120-124: The same is repeated twice.

Lines 136-148: The same is repeated twice.

Fig 2A: Should include the "untreated" control. If we compare the "untreated" signals of other histograms, we can see that a fraction of the toxin (up to 30%) at 4ºC is internalized or is being protected from trypan blue quenching. What can you comment about that?
